# Fuel Pretreatment Systems in Modern CI Engines

**Jacek Eliasz [1], Tomasz Osipowicz [1,*], Karol Franciszek Abramek [1] , Zbigniew Matuszak [2]
and Łukasz Mozga [1]**

[1] Department of Automotive Engineering, West Pomeranian University of Technology,
70-311 Szczecin, Poland; Jacek.Eliasz@zut.edu.pl (J.E.); Karol.Abramek@zut.edu.pl (K.F.A.);
lukasz.mozga@zut.edu.pl (Ł.M.)

[2] Faculty of Marine Engineering, Maritime University of Szczecin, 71-650 Szczecin, Poland;
z.matuszak@am.szczecin.pl

* Correspondence: tosipowicz@zut.edu.pl; Tel.: +48-503420350

**Abstract:** The article concerns the possibility of using a fuel pretreatment system in modern compression ignition CI engines, the main task of which is the reduction of toxic emissions in the form of exhaust gases. This fuel pretreatment system consists of a catalytic reactor used in common rail (CR), and a modified fuel atomizer into spiral-elliptical channels covered with catalytic material. In the system presented here, platinum was the catalyst. The catalyst's task is to cause the dehydrogenation reaction of paraffin hydrocarbons contained in the fuel to create an olefin form, with the release of a free hydrogen molecule. In the literature, the methods of using catalysts in the exhaust systems of engines, or in combustion chambers, injection pumps, or fuel injectors, are known. However, the use of a catalytic reactor in the CR system in a high-pressure fuel atomizer rail is an innovative project proposed by the authors. Conditions in the high-pressure CR system are favorable for the catalyst's operation. In addition, the spiral-elliptical channels made on the inoperative part of the fuel atomizer needle increase the flow turbulence and contact surface for the catalyst.

**Keywords:** CI engine; high-pressure rail; fuel injector; fuel processing

## 1. Introduction

Modern engines with self-ignition are subject to environmental protection concerns. The reduction of pollution in exhaust gases is achieved through the use of particulate filters and oxidation catalysts in exhaust systems, the use of electronic injection systems, fuel additives, and alternative fuels. This paper proposes an innovative design for a fuel pretreatment system in the common rail (CR) system. The purpose of the changes to the fuel supply system proposed by the authors is to improve the combustion process and reduce the emission of toxic substances in the exhaust gas in a modern CI engine by using spiral-elliptical channels made on a nonworking part of the fuel atomizer needle. During the implementation of the project, an analysis of the literature was carried out, and preliminary studies were carried out to determine the direction of further analysis.

The use of a fuel pretreatment system in the form of a catalytic reactor in a CR high-pressure accumulator and spiral-elliptical channels made on a nonworking needle covered by a platinum catalyst is an innovative concept proposed by the authors. Earlier research conducted by the authors [1] showed that the application of a platinum catalyst on the inoperative part of the needle by the electrospark method slightly reduced the emission of toxic substances into the atmosphere, and the fuel consumption of the ZS engine. It should be noted, however, that the tests were conducted on an old type of engine with mechanical direct fuel injection. Gianotti et al. [2] discuss the method of partial dehydrogenation of hydrocarbons contained in fuels in order to obtain pure hydrogen. Platinum on an aluminum medium was used as a catalyst. Research results indicate that it is possible to cause a

dehydrogenation reaction of hydrocarbons contained in the fuel with the release of hydrogen molecules. The authors of [3] examined the possibility of the dehydrogenation of cyclohexane to the form of cyclohexene. The analysis showed a high efficiency of the platinum catalyst in dehydrogenation reactions. The theoretical analysis presented in [4] shows that platinum catalysts are used for the dehydrogenation reaction of various chemical compounds. Catalyst activity increases as the platinum content increases. High temperature and pressure are the conditions in a CR high-pressure system. Such an environment is favorable for the initiation of reactions involving catalysts. The process of atomizing fuel in CI engines is very important for the formation of a combustible mixture in the combustion chamber. Engine operating and ecological parameters depend on the quality of the fuel jet, which is influenced by the atomizer geometry. In [5], five types of single-hole cylindrical injectors with different injection port diameters (0.13–0.23 mm) and their lengths (0.7–1.0 mm) were analyzed. The research results showed that the diameter of the Sauter droplets (SMD) decreases with increasing distance from the atomizer, and is larger in the stream core than at its periphery. In addition, it decreases with the increasing pressure in the SMD system. The diameter and length of the injection holes slightly affect the droplet size in the stream. Engine tests have shown that SMD fuel drops affect the combustion process. Smaller drops improve the heat release process during combustion. In [1], the process of diesel fuel injection and its mixtures with gasoline in the common rail system were assessed. Macroscopic parameters of injection, such as opening angle, stream width, penetration degree, and SMD diameter of droplets were examined. The test results showed that the mixture of diesel and gasoline caused a shorter delay in self-ignition, earlier fuel injection time, and longer duration. SMD was smaller for a mixture of diesel and gasoline, which improved the atomization of the droplets in the stream. Similar results were obtained by other researchers [6]. In addition, the process of atomizing the fuel mixture with ethanol was investigated. Based on the results of the analysis, it was found that the addition of ethanol increased the penetration of the stream and reduced its angle of opening. The mixture of diesel and gasoline resulted in smaller drop diameters than with the addition of ethanol. The phenomenon of cavitation plays an important role during the fuel flow through the atomizer. In [7], the development of air bubbles in a transparent atomizer was examined. Direct visualization of the follicular boiling process was carried out using a high-resolution microscope and a specialized camera. The analysis showed that increasing the fuel temperature causes cavitation to increase, which causes a larger angle of opening, penetration, and dispersion of the stream. The phenomenon of cavitation affects these parameters. Similar research results were obtained in [5]. The analysis showed that air bubbles inside the atomizer and right at its outlet affect the stream quality parameters. The scale of the bubble boiling phenomenon increases with increasing fuel temperature. Analyses of the influence of temperature on the operating parameters of common rail system injectors were carried out in [8,9]. Studies have shown that fuel temperature influences injection rates, especially at low temperatures. The authors of [10] introduced the K factor, which determined the atomic geometrical parameters. This depends on the diameter of the injection hole at the inlet and outlet, as well as its length. The studies have shown that, with the increase of the coefficient K of the atomizer, the penetration ability of the stream was also increasing, but its angle of opening was decreasing. An important role in the combustion process is played by physical fuel parameters, which influences the injection time, angle, dosage, delay, and nozzle open pressure. These factors affect the fuel stream properties. The fuel stream enters into the equation of carbon monoxide and soot in fume formation. A very fast increase in the pressure in the combustion chamber induces higher emissions of nitrogen oxides. Fuel parameters do not affect the Reynolds number, only the flow in the nozzle. A CFD simulation of the fuel injection process was conducted in two stages: incompressible flow at constant liquid properties and compressible flow at liquid properties calculated locally as a function of flow pressure conditions [11]. Issues related to fuel flow through the injector atomizer should be considered in terms of the compressibility of the liquid, depending on the pressure and flow temperature. At low system pressures, the flow is moderate, and the liquid is in a state of transition between laminar and turbulent flow. The injection dose factor is then dependent on the Reynolds number, but when the pressure in the system begins to increase,

the flow goes into the turbulent phase, and the injection dose factor is independent of the Reynolds number [11–13].

Tests carried out on the injection and combustion process in diesel engines are carried out using different methods. However, the use of the catalytic system in high-pressure rail and the fuel injector atomizer are innovative elements proposed by the authors of this paper.

## 2. Fuel Pretreatment Systems in Modern CI Engines

Fuel pretreatment is improving the combustion process in modern CI engines. Considering the structure of the common rail system, it is possible to install an additional catalytic reactor outside the fuel injector in the high-pressure battery–(see Figure 1) [14]. A reactor was installed inside the rail: a rod with annular channels and a platinum catalyst. The purpose of the channels is to increase the contact surface for the catalyst and to induce initial fuel turbulence. The second element of the system modified by the authors are fuel injector atomizers (see Figure 2) [15]. Spiral-elliptical channels were made on the nonworking part of the needle, and the catalytic effect of platinum was applied to them. The task of the channels is to cause additional turbulence of fuel flow through the atomizer and to increase the contact surface of the catalyst. It is an innovative solution proposed by the authors of this article.

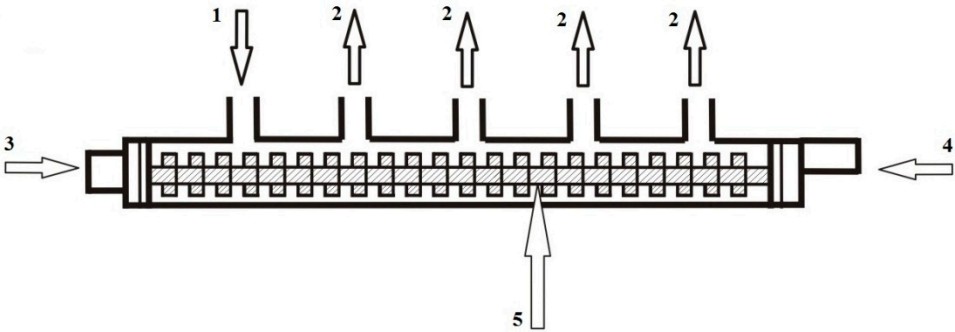

**Figure 1.** Modified common rail (CR) high-pressure battery: 1—fuel supply to the rail, 2—output for fuel injectors, 3—high-pressure sensor, 4—high-pressure regulator, 5—catalytic reactor.

The process of combustion and heat generation plays a fundamental role during the work of the compression-ignition engine. The main characteristics of this phenomenon are the time and speed of the pressure and temperature rise in the engine compartment. An important role in shaping the phenomena associated with the combustion of combustible mixtures in diesel engines is the period of self-ignition delay. This is the time between the start of fuel injection into the combustion chamber of the engine and the appearance of the first focal point of self-ignition. On the indicator chart, it is recorded as a rapid increase in temperature and pressure. The theory of combustion aims to maximize this time. A detailed analysis of the combustion process of combustible mixtures in engines is discussed in [16]. The auto-ignition delay ($\tau_i$) of the combustible mixture can be calculated from the experimental data of N. N. Semonow according to Equation (1):

$$\tau_i = const + \frac{T^m}{p^n} e^{\frac{E_a}{RT}} \tag{1}$$

Equation (1) shows that the auto-ignition delay period depends on the temperature and pressure in the engine's combustion chamber and on the value of the activation energy *Ea*.

The activation energy is the basic value determining the course of a chemical reaction. By reducing its value, we increase the constant reaction speed, so it occurs faster. Physically, the action of activation energy has been described in detail in [17], and can be explained in the following way. During the reaction, collisions of reactive particles occur. Effective collisions are those whose energy at the moment of collision is greater than the average energy determined for a given temperature. Activation energy is

the excess energy that particles should have at the moment of collision to react [17]. Hydrocarbon fuels are multi-atomic systems, so activation energy can be defined as the minimum kinetic energy, which should be greater than the potential energy of the system for a chemical reaction to occur. To sum up, it is an energy barrier that substrates have to overcome to enter the product phase. By reducing its value, we shorten the auto-ignition delay period, which improves the combustion process in the engine compartment. CI engine fuels mainly consist of $CnH_{2n+2}$ paraffin hydrocarbons. In these hydrocarbons, the breaking energy of C-H bonds is greater than the energy of C-C bonds; therefore, with the increase in carbon atoms, less activation energy is needed to break the molecule [17]. Providing more energy to the reaction environment, e.g., by heating or using a substance (catalyst) that reacts with the substrate, means the resulting compound passes faster into the final product, which will help overcome the energy barrier. It is possible to change the chemical properties of the fuels used in CI engines by dehydrogenating the most numerous group of paraffin hydrocarbons in the presence of a catalyst to the $CnH_{2n}$ olefin group with the separation of a hydrogen molecule. Hydrogen, thanks to its high diffusion coefficient, high ignition capacity, combustion rate, and wide flammability limits, helps to reduce the autoignition period under the conditions of the combustion chamber. In addition, the presence of a hydrogen molecule in the atomized fuel stream, due to its high diffusion coefficient, can accelerate the process of evaporation and the mixing of fuel with air [1]. The catalysis phenomenon involves exchanging one (higher) activation energy (without catalyst) for two or more smaller activation energies. This means that the route from substrates to products changes the elementary reaction sequence with a catalyst. There are low activation energies in particular phases (Figure 3). The process proceeds by way of indirect products, involving the desired products and the reconstruction of the catalyst, so, during the whole process, the catalyst is conserved.

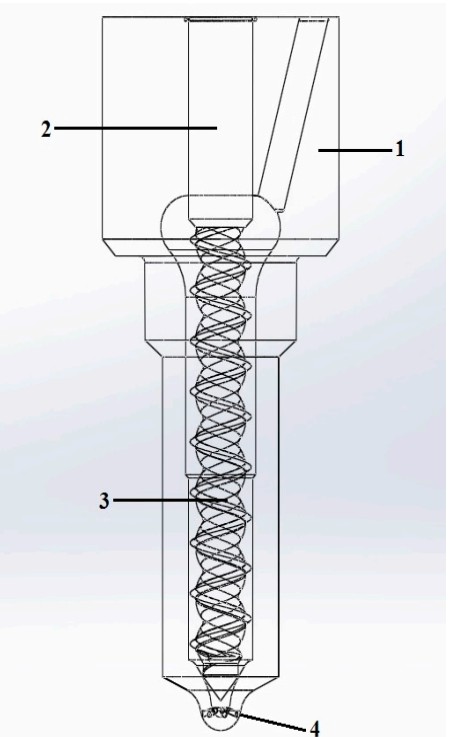

**Figure 2.** Modified fuel injector atomizer: 1—case, 2—precise needle pair, 3—spiral-elliptical channels made on the inoperative part of the needle with a catalytic reactor, 4—injection holes.

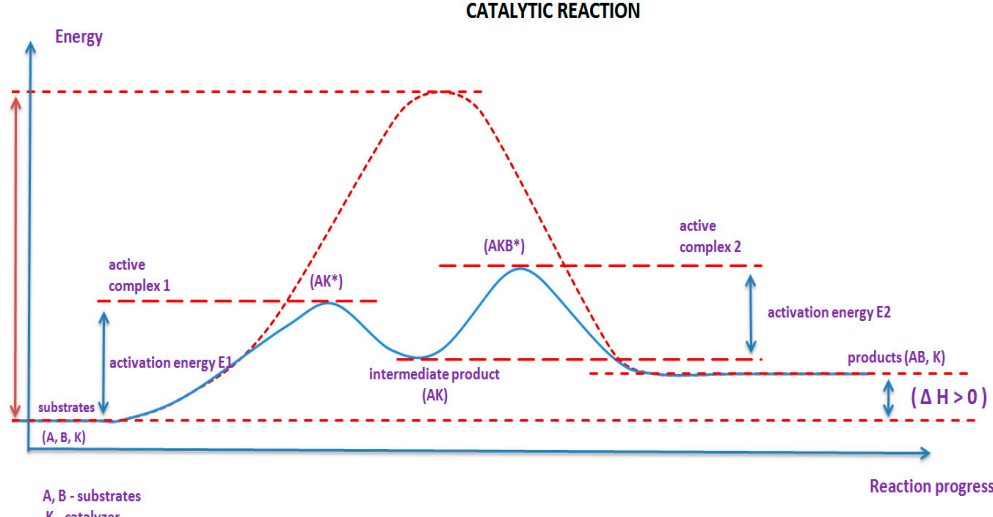

**Figure 3.** Course of chemical reaction with catalyst.

We can accelerate reactions by heterogeneous catalysts in a few ways:

- Substrates are transferred from liquid or gas phase to the catalyst surface; this stage is very slow and controlled by the diffusion speed. This phase could be controlled by moderating the speed (turbulence);
- Substrates are absorbed onto the catalyst surface, controlled by the absorption speed,
- Intermolecular substrates formed in reactions are absorbed onto the catalyst surface; this process is controlled by the surface reaction speed;
- Reaction products are desorbed from the catalyst surface to the phase interior; this stage is controlled by the desorption speed;
- Reaction products are transported from the catalyst surface to the phase interior; this stage, like the first stage, is controlled by the diffusion speed.

The task of the catalytic reactor mounted on the CR rail is initiating the fuel dehydrogenation reaction before it flows to the injectors. The spiral-elliptical channels made on the inoperative part of the atomizer needle, in addition to increasing the contact area with the catalyst, cause additional turbulence in the fuel flow. A feature of turbulent flows is fluctuation in the momentum, kinetic energy, and the phenomenon of heat exchange. The physical fuel parameters, such as viscosity and density, depend on the temperature. If the local fuel temperature in the atomizer increases as a result of additional vortices, then the physical parameters of the fuel will also change, which will affect the atomization process. The fuel pretreatment system in the common rail system is an innovative concept proposed and presented by the authors, which has not yet been presented anywhere else.

## 3. Simulation Studies' Results

Simulation studies were carried out in the Solidworks Flow Simulations environment. The aim of the analysis was to investigate how the fuel moves in a standard and modified fuel atomizer, and how the temperature distributes during the flow. It should be noted that this simulation is just an overview, and its task is to visualize the processes taking place in the fuel injector atomizer.

The simulations have been carried out according to the equations of mass, momentum, and energy [18], where $u$ is a fluid speed, $\rho$ is the density, $S_i$ is the external mass forces, $g_i$ is the standard gravity, h is the thermal enthalpy, $Q_H$ is the source of heat, $\tau_{ij}$ is the stress-velocity tensor, $q_i$ is the heat

flux, $\Omega$ is the angular velocity of the whirl, $r$ is the rotation radius, $k$ is the turbulence kinetic energy, and $h_m{}^0$ is the individual thermal enthalpy:

$$\frac{\partial \rho}{\partial t} + \frac{\partial}{\partial x_i}(\rho u_i) = 0 \tag{2}$$

$$\frac{\partial \rho u_i}{\partial t} + \frac{\partial}{\partial x_j}(\rho u_i u_j) + \frac{\partial p}{\partial x_i} = \frac{\partial}{\partial x_j}\left(\tau_{ij} + \tau_{ij}^R\right) + S_i\ i = 1,\ 2,\ 3 \tag{3}$$

$$\frac{\partial \rho H}{\partial t} + \frac{\partial \rho u_i H}{\partial x_i} = \frac{\partial}{\partial x_i}\left(u_j\left(\tau_{ij} + \tau_{ij}^R\right) + q_i\right) + \frac{\partial p}{\partial t} - \tau_{ij}^R\frac{\partial u_i}{\partial x_j} + \rho\varepsilon + S_i u_i + Q_H \tag{4}$$

$$H = h + \frac{u^2}{2} + \frac{5}{3}k - \frac{\Omega^2 r^r}{2} - \sum_m h_m^0 y_m \tag{5}$$

$$q_i = \left(\frac{\mu}{Pr} + \frac{\mu_t}{\sigma_c}\right)\frac{\partial h}{\partial x_i},\ i = 1,\ 2,\ 3 \tag{6}$$

Knowing the flow rate and flow geometry (injection dose and flow field), the velocity of the fuel in the atomizer can be calculated on the basis of the following equation:

$$V_1 = \frac{\dot{Q}}{A} \tag{7}$$

Knowing the $V_1$ value from Bernoulli's equation for fuel velocity, $V_2$ at the atomizer's discharge can be calculated as:

$$V_2 = \sqrt{V_1^2 + \frac{2p_2 - 2p_1}{\rho}} \tag{8}$$

Pressure values in the injection system ($p_1$) and combustion chamber ($p_2$) are known. After determining the fuel flow velocity in the atomizer and at the outlet of the nozzle, the Reynolds number can be calculated:

$$R_e = \frac{\rho V d_0}{\mu} \tag{9}$$

where $V$, $V_1$, and $V_2$ are the flow velocities, $p_1$ is the pressure in the injector, $p_2$ is the fuel pressure at the outlet of atomizer's nozzle, $Q$ is the mass flow, A is the flow surface in the atomizer, $\rho$ is the fuel density, $\mu$ is the kinematic viscosity, and $d_0$ is the flow value.

Figure 4 presents the influence of pressure in the system on the fuel speed in the atomizer. As can be observed, the higher the pressure in the system, the higher the flow speed of the fuel in the atomizer. Figures 5 and 6 present a simulation of fuel flow through the modified high-pressure accumulator of the common rail system.

Figures 7–9 present the simulation results of fuel flow through the atomizer with fluid temperature distribution.

Figure 10 presents the temperature distribution of the atomizers' frames, standard (2) and modified (1), on the bench during their operation. The modified fuel atomizer (1) achieved a higher operation temperature than the standard one (2).

During laboratory tests on the bench, the fuel temperature at the atomizer's discharge was measured. These measurement results are shown in Figure 11.

Figure 12 shows how the fuel discharge temperature changes, depending on the pressure in the injection system.

Engine studies were conducted as part of the initial analysis to see if there is a possibility of improving the eco-parameters of the self-ignition engine with the common rail system by applying the innovative changes in the engine driving system proposed by the authors. During the standard measurement of fuel injection doses for a pressure of 135 MPa, the distribution of temperatures on

the atomizers, standard and modified, has been examined according to [19]. We also performed a visualization of the stream of injected fuel as well as studies on the engine test beds for the engine Fiat 1.3 JTD, during which the emission of nitrogen oxides, carbon oxides, and smoke was measured.

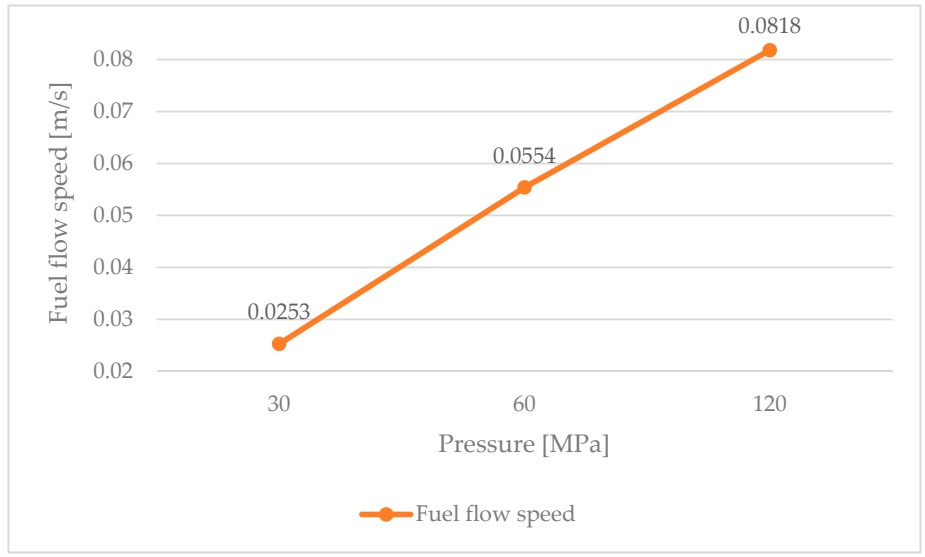

**Figure 4.** Fuel flow speed inside the injector nozzle depends on the system pressure [19].

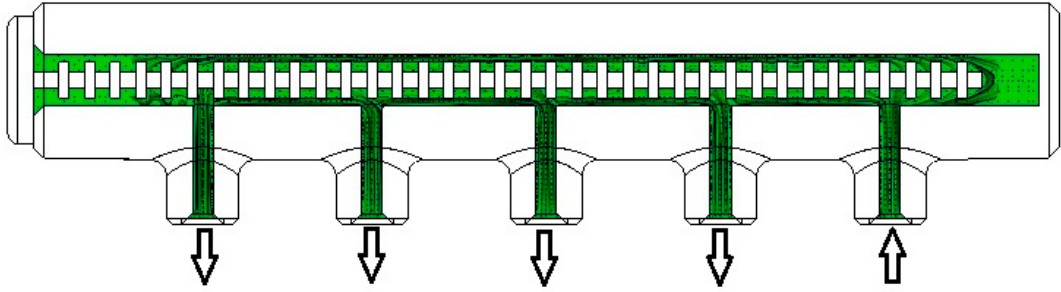

**Figure 5.** Simulation of fuel flow through CR.

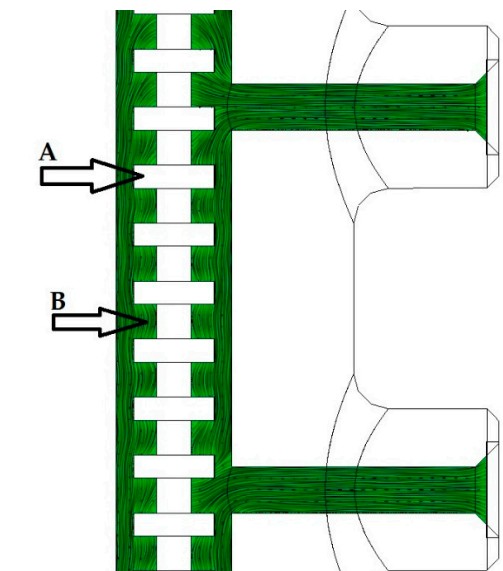

**Figure 6.** Common rail with catalyst. (**A**) Catalyst reactor; (**B**) fuel flowing through the reactor.

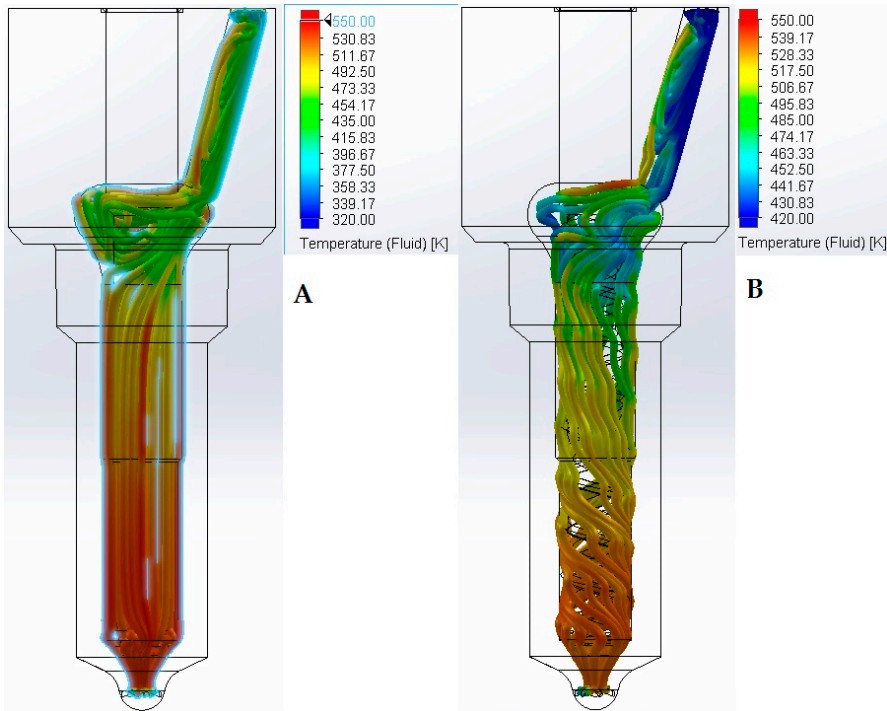

**Figure 7.** The results of preliminary simulation studies of the flow and temperature distribution of the fuel through the standard and modified atomizer. Flow parameters: injection dose: 35.3 mm³/H, injection time: 780 μs, pressure in the system: 135 MPa. (**A**) Standard atomizer; (**B**) modified atomizer.

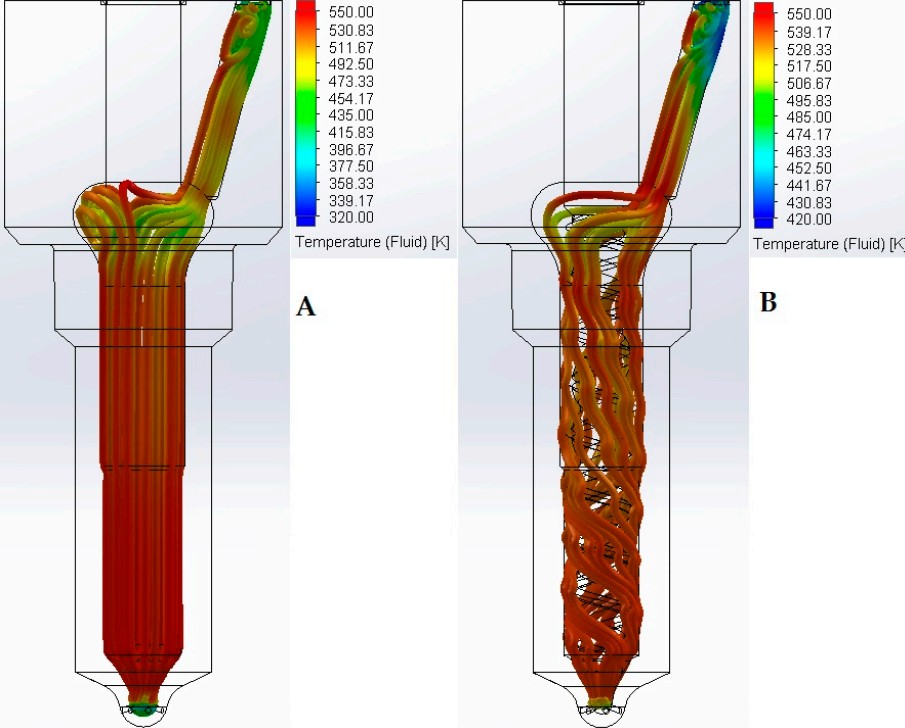

**Figure 8.** The results of preliminary simulation studies of the flow and temperature distribution of the fuel through the standard and modified atomizer. Flow parameters: injection dose: 0.7 mm³/H, injection time: 420 μs, pressure in the system: 30 MPa. (**A**) Standard atomizer; (**B**) modified atomizer.

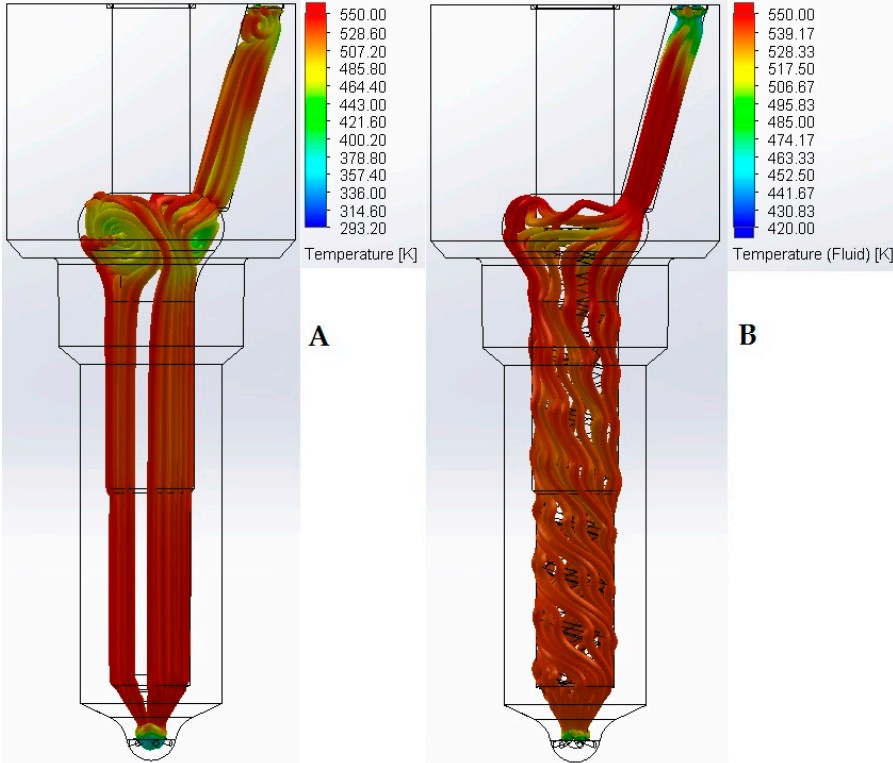

**Figure 9.** The results of preliminary simulation studies of the flow and temperature distribution of the fuel through the standard and modified atomizer. Flow parameters: injection dose: 1.2 mm$^3$/H, injection time: 260 μs, pressure in the system: 80 MPa. (**A**) Standard atomizer; (**B**) modified atomizer.

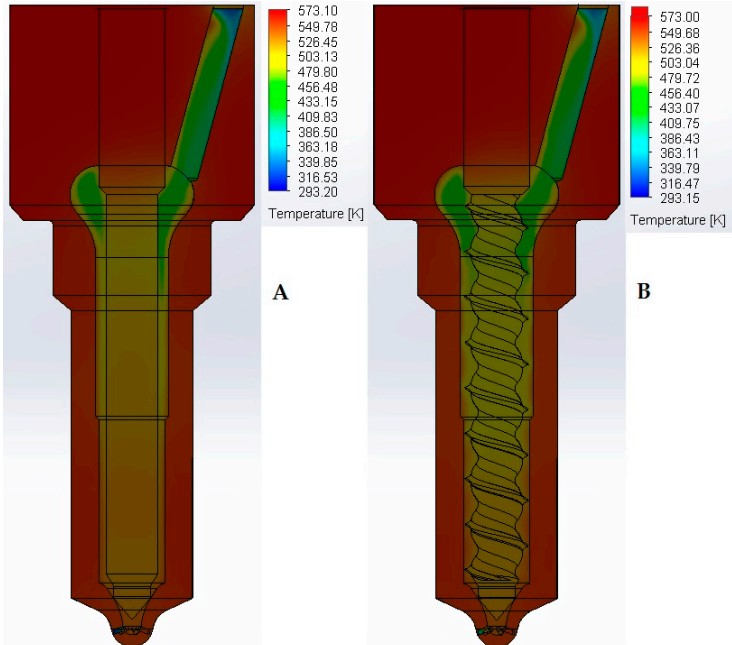

**Figure 10.** The results of preliminary simulation studies of temperature distribution of body and needle at the pressure of 135 MPa. (**A**) Standard atomizer; (**B**) modified atomizer.

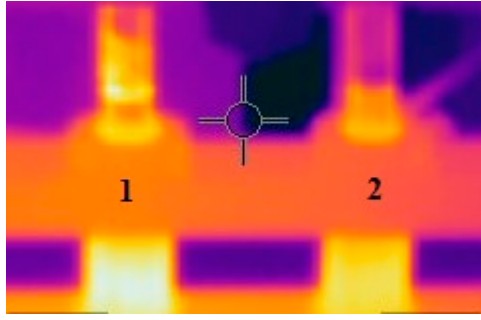

**Figure 11.** Temperature distribution tests in atomizers: (**1**) modified; (**2**) standard.

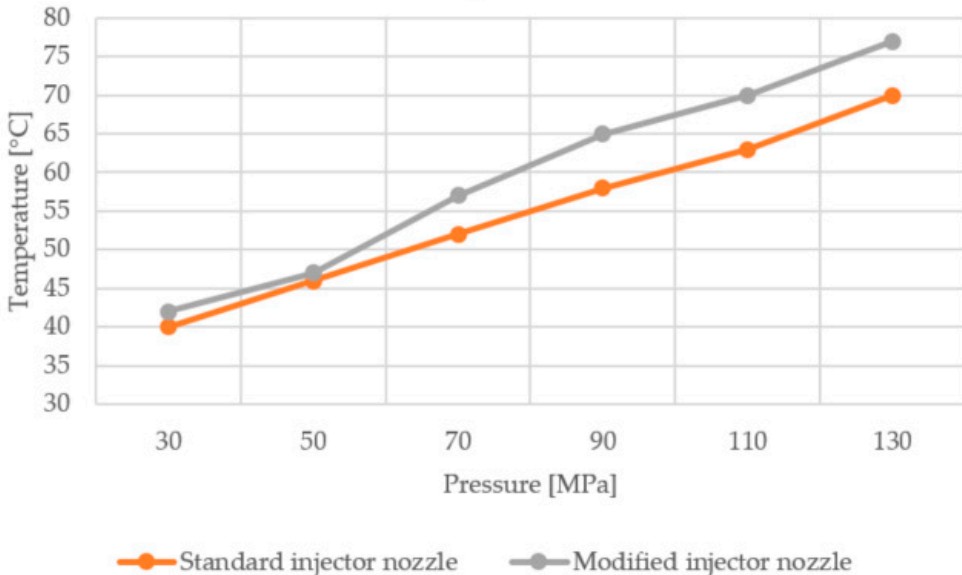

**Figure 12.** Fuel temperature results at the discharge from the atomizer with different pressures in the system.

Figures 13–15 show the results of the measurement of nitrogen oxides, carbon monoxide, and exhaust fumes for an engine with a standard and modified common rail system. The research object was a Fiat 1.3 JTD engine with a CR system.

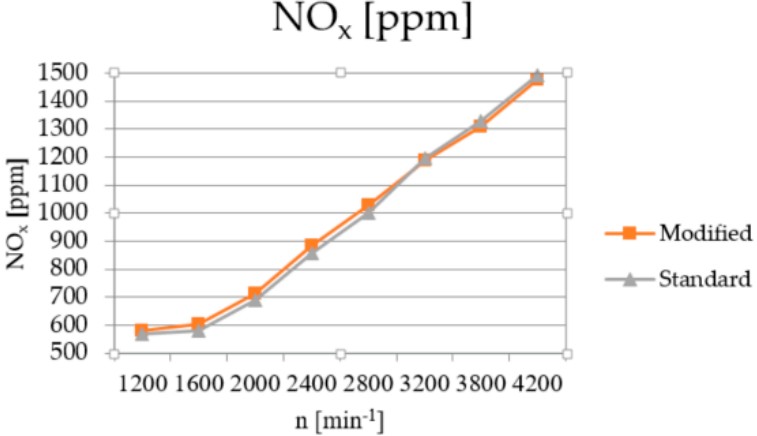

**Figure 13.** Results of nitrogen oxide $NO_x$ measurements for the standard CR system and the modified CR system.

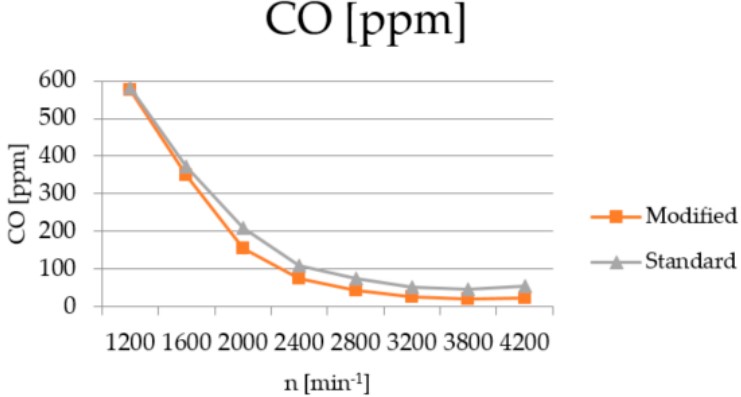

**Figure 14.** Carbon monoxide CO measurement results for the standard CR system and the modified CR system. Measurement error of exhaust fumes analyzer 1.5%.

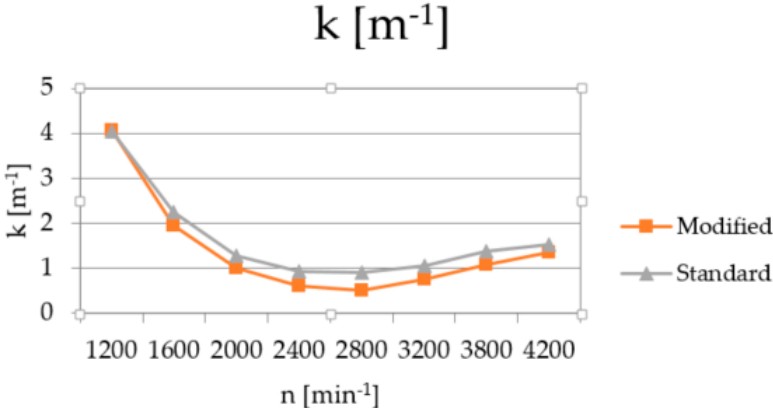

**Figure 15.** Exhaust fumes results for the standard CR system and the modified CR system Measurement error of the opacimeter: 2%.

## 4. Discussion

Applying a fuel pretreatment system to self-ignition engines with the common rail system is an innovative project. The theoretical analysis of the use of the catalyst in the CR rail and the atomizer indicated that it is possible to cause a dehydrogenation reaction of paraffin hydrocarbons, converting them to olefinic form, with the release of a free hydrogen molecule. The fuel supply system has pressures from 20 to about 200 MPa. The fuel temperature in the rail reaches 330 K and, in the atomizer just before the injection process, up to 500 K. Such conditions favor the operation of the catalyst. Additionally, the spiral-elliptical channels increase the contact surface of the catalyst and cause additional vortices that change the nature of the fuel flow through the atomizer (see Figures 7–9). A simulation of the temperature distribution of the atomizer's body and needle is presented in Figure 12. The studies have shown that the temperatures of both atomizers are similar, but that the fuel temperature in the modified atomizer is higher. It is affected by the additional vortices, thanks to which the fuel additionally flows around the needle and heats up. Analysis of the temperature distribution of modified and standard atomizers during operation with a thermal imaging camera is presented in Figure 11. The calculations have shown that the modified atomizer generates a higher temperature during work than a standard one, which confirms the relevance of the simulation carried out by the authors. The increase in the fuel temperature of the atomizer is caused by the fact that the fuel flowing around the needle through the channels stays there longer and heats up further, which affects the qualitative parameters of the injected fuel stream, such as range, angle of aperture, atomization, and surface and outlet velocity. Improving these factors causes faster evaporation and mixing of fuel with air.

The aim of fuel pretreatment is to improve the eco-parameters of the modern CI engine. The task of the catalysts installed in common rail as well as in fuel atomizer is to initiate a dehydrogenation reaction of paraffin hydrocarbons to olefin with the separation of free hydrogen particles.

$$\text{catalyst} \quad \Downarrow \Downarrow \quad \text{temperature}$$

$$C_nH_{2n+} \longrightarrow C_nH_{2n}+H_2 \tag{10}$$

The presence of hydrogen in the injected fuel may influence the combustion process and improve the physical parameters of the fuel [20]. Engine tests presented in Figures 13–15 have shown that the emissions of nitrogen oxides remained at the same level for the standard and modified systems. Carbon monoxide and smoke emissions were slightly reduced for engines with modified power systems, especially at higher engine speeds. This could have been influenced by the modification of the fuel atomizer and the pressure in the system. Pressure in the system affects the fuel velocity at the outlet to the combustion chamber. At higher pressures in the system, fuel velocity at the outlet to the combustion chamber increases, which affects the stream parameters. Therefore, additional turbulence could lead to lower carbon oxide and soot emissions by the engine through better fuel atomization.

At a lower engine load, when the pressure in the system decreases, the liquid speed in the atomizer drops. The fuel flows more slowly through the channels and heats up from them (Figure 10). Higher fuel temperature with additional hydrogen particles can affect the self-ignition delay period, making it shorter, and improving the combustion process.

## 5. Conclusions

From the conducted laboratory and engine preliminary tests, it can be stated that the implementation of the fuel pretreatment system in the common rail system is a worthwhile project.

In order to conduct a comprehensive analysis of the possibilities of using the pretreatment system in common rail systems in the second stage of project research, the below suggestions should be considered:

1.  Carry out laboratory tests on the injected fuel stream with standard and modified atomizers. During the tests, a qualitative stream analysis will be performed.
2.  Carry out engine tests during which the operating and ecological parameters of the engine will be measured, and the measurement of fast-changing pressures and temperature will be carried out using a standard and modified fuel supply system.

**Author Contributions:** The paper was written based on the patents by T.O., Patent No. 222791; T.O. and K.F.A. Patent Application No. 233791; and T.O. and K.F.A., Patent No. 234823. J.E., Z.M., K.F.A. and Ł.M. carried out the analysis. T.O. wrote the paper. All authors have read and agree to the published version of the manuscript.

**Funding:** This research received no external funding. The APC was funded by Maritime University of Szczecin.

**Conflicts of Interest:** The authors declare no conflict of interest. The funders had no role in the design of the study; in the collection, analyses, or interpretation of data; in the writing of the manuscript, or in the decision to publish the results.

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
