# Peer review of "Fuel Pretreatment Systems in Modern CI Engines"

_catalysts, doi:10.3390/catal10060696_

Round 1
Reviewer 1 Report
This manuscript is very similar to another work of these authors [Eliasz, J., Osipowicz, T., Abramek, K. F., & Mozga, Ł. (2019). Model Issues Regarding Modification of Fuel Injector Components to Improve the Injection Parameters of a Modern Compression Ignition Engine Powered by Biofuel. Applied Sciences, 9(24), 5479].
The novelty of this work should be determined.
Author Response
The article Model Issues Regarding Modification of Fuel Injector Components to Improve the Injection Parameters of a Modern Compression Ignition Engine Powered by Biofuel describes the possibility of of using spiral-elliptical channels on not working part of fuel injector needle modern CI engine. There has been made fuel flow model and analysed phenomenom in injector nozzle. Based on literature analysis and fuel flow models described in the previous paper there has been used platinum catalyst in fuel injection system modern CI engine. The article describes the meaning of using and working catalysts in fuel injection systems. Simulations show that fuel injector nozzle is the good place to install catalytic layer. The spiral – elyptical channels described in article published on Applied Sciences increasing contact surface catalyst and fuel. Flowing fuel round the channels undergoes heating, what favor catalyst work. The second place where catalyst can be install is high pressure rail. Temperature and pressure inside rail is good environment for platinum catalyst. The article shows initially engine researches, where carbon monoxide, solid particles and nitric oxides has been measured. Initially researches presents small differences between standard and modified fuel injectors, but makes a good starting point for second stage of researches. We think that modified fuel injectors should be adjust. There is planed second part of engine research by using engine indicator sensor. It shows the differents of pressures and temperatures in engine combustion chamber with standard and modified fuel injectors, and helps by adjustment.

Reviewer 2 Report
Dear Authors. Thnak you for the respοnse, More effortsmust be made in order to describe adequately the work of the catalytic reactor and its hydrogenation process before the common rail system. Its input assumed to be critical but the evidence why is critical must be described accordingly.
Editorial comments is the abbreviation must be made visible when it is apperared for the first time.
Author Response
Catalytic treatment uses in modern injection system is an innovative conception. Initially researches and simulations show that it is possible to improve ecological parameters in modern CI engines using platinum catalyst in fuel injection system The main taks of initially engine researches was analysis engine work on modified fuel injectors. The results show that carbon monoxide and solid particles has been lowered, but nitrogen oxides emission were on the same level. We think that modified fuel injectors should be adjust. There is planed second part of engine research by using engine indicator sensor. It shows the differents of pressures and temperatures in engine combustion chamber with standard and modified fuel injectors, and helps by adjustment.

Round 2
Reviewer 1 Report
Sorry, I don't understand the difference between this article and this article.
Author Response
The article in Catalyst concerns the influence of using catalytic layer on not working part of fuel injector needle and high pressure rail to make fuel pre-treamtment phenomena before injection process. The aim of fuel pre-treatment system is improvement CI engines ecological parameters. To install catalyst in injector needle used the channels described in article published in Applied Sciences, because of increases surface for catalytic layer and fuel temperature during flow. The paper “Model Issues Regarding Modification of Fuel Injector Components to Improve the Injection Parameters of a Modern Compression Ignition Engine Powered by Biofuel” describes earlier researches with using only channels on not working part of needle without catalyst. The difference between article is that there has been made analysis the phenomena concerning fuel and biofuel flow through spiral-elliptical ducts. We have noticed that there is possibility to put on this ducts catalyst to make fuel dehydrogenation reaction. There has been performed platinum catalytic converter in high pressure rail (patent no. 234823) and in the injector nozzle. The difference between articles is that in Catalyst has been evaluated influence the catalytic layer on toxic substances emission in modern CI engine. This solution has been never used in engines with Common Rail system.
This manuscript is a resubmission of an earlier submission. The following is a list of the peer review reports and author responses from that submission.
Round 1
Reviewer 1 Report
Upon reviewing this paper, I find the quality of the overall research to be very poor and I highly recommend that this paper be rejected on the basis of technical merit and lack of new, unique content,and English grammar. Research background and analysis of results are very poor. The authors only gave a simple and low description on the data in figures 3 to 7, and seriously lack of in-depth analysis and comparative analysis using related some literatures.
Author Response
This article presents theoretical analysis of using possibilities catalyst in the fuel injectors components engine with Common Rail system. We have added the thermal flow simulations and initial fuel stream visualisation.
We prepare engine test bench. The second stage of researches will be made with pressure measure in the combustion chamber and presented in the second part of the paper.
English quality depends on our technical translator. We sent the paper to make corrections.
Reviewer 2 Report
The authors evaluated a new design of the fuel pre-treatment system in the common rail process, which is reduced the emission of toxic substances in exhaust gases. Overall I think the paper is interesting (within the scope of this journal), but there are quite a number of major revisions that need to be done. Some considerations which must be addressed by the authors: The novelty of the manuscript. Really is very confuse. This must be remarked in the article. The novelty of the work is not clear. Provide the comparative study by comparing your results with other reported results which will highlight the importance your work. In addition, authors published some related work with this subject in other conferences and journals, they need to clarify the difference of this work with their previous works. Style of the writing of this research (e.g. introduction and literature review) different from the standard article. This style may be suitable for conference paper but it is not acceptable for journal paper.
Author Response
This article presents theoretical analysis of using possibilities catalyst in the fuel injectors components engine with Common Rail system.
The style and introduction has been corrected. We have added the thermal flow simulations and initial fuel stream visualisation.
English quality depends on our technical translator. We sent the paper to make corrections.
Reviewer 3 Report
The paper entitled: "The concept of using the fuel pretreatment system in modern CI engines." is tryig to examine the use of a catalytic convereterin the fuel pretreatment system for CI engines. The paper must be improved. The presentation of the diagrams must be improved including error analysis. The results of the simulation analysis must be presented providing more scientific details.
English language presentation must be improved (e.g. Research conducted by the authors [6]..., The paper [4] discusses...., The authors of the paper [15]... etc.)
Author Response

(The authors gave the same response as above.)

Round 2
Reviewer 1 Report
The authors did not revise this manuscript according to my requirements, and this article still lacks in-depth analysis.
Therefore, I do not recommend that the current format of this article be published in Catalysts.
Reviewer 2 Report
The structure of the manuscript is not standard. The conclusion part is not standard, it's better some of the paragraphs move to results part.
Reviewer 3 Report
Dear authors of the manuscript entitled "The concept of using the fuel pretreatment system in modern CI engines" please take notice of the following observations. The connection between the simulation study and the experimental part is insufficient. How the experimental results support the findings. The experimental apparatus with the modifications in the FIAT engine must be clearly presented.
The improvements must be accompanied by sufficient experimental results
The measurements of emissions seems to be with no connection with the previous "body" of the document. Which are:
the analytical technics? the fuel properties? the analyzers? the error analysis of the results?